# The Yin-Yang Pharmacomicrobiomics on Treatment Response in Inflammatory Arthritides: A Narrative Review

**DOI:** 10.3390/genes14010089

**Published:** 2022-12-28

**Authors:** Silvia Peretti, Sara Torracchi, Edda Russo, Francesco Bonomi, Elisa Fiorentini, Khadija El Aoufy, Cosimo Bruni, Gemma Lepri, Martina Orlandi, Maria Sole Chimenti, Serena Guiducci, Amedeo Amedei, Marco Matucci-Cerinic, Silvia Bellando Randone

**Affiliations:** 1Department of Clinical and Experimental Medicine, University of Florence, Largo Brambilla 3, 50134 Florence, Italy; 2Department of Rheumatology, University Hospital of Zurich, University of Zurich, 8006 Zurich, Switzerland; 3Rheumatology, Allergology and Clinical Immunology, Department of Medicina dei Sistemi, University of Rome Tor Vergata, 00133 Rome, Italy; 4Unit of Immunology, Rheumatology, Allergy and Rare Diseases (UnIRAR), IRCCS San Raffaele Hospital, 20132 Milan, Italy

**Keywords:** pharmacomicrobiomics, gut microbiota, inflammatory arthritides, personalized medicine

## Abstract

(1) Background: Gut microbiota (GM) is the set of microorganisms inhabiting the gastroenteric tract that seems to have a role in the pathogenesis of rheumatic diseases. Recently, many authors proved that GM may influence pharmacodynamics and pharmacokinetics of several drugs with complex interactions that are studied by the growing field of pharmacomicrobiomics. The aim of this review is to highlight current evidence on pharmacomicrobiomics applied to the main treatments of Rheumatoid Arthritis and Spondyloarthritis in order to maximize therapeutic success, in the framework of Personalized Medicine. (2) Methods: We performed a narrative review concerning pharmacomicrobiomics in inflammatory arthritides. We evaluated the influence of gut microbiota on treatment response of conventional Disease Modifying Anti-Rheumatic drugs (cDMARDs) (Methotrexate and Leflunomide) and biological Disease Modifying Anti-Rheumatic drugs (bDMARDs) (Tumor necrosis factor inhibitors, Interleukin-17 inhibitors, Interleukin 12/23 inhibitors, Abatacept, Janus Kinase inhibitors and Rituximab). (3) Results: We found a great amount of studies concerning Methotrexate and Tumor Necrosis Inhibitors (TNFi). Conversely, fewer data were available about Interleukin-17 inhibitors (IL-17i) and Interleukin 12/23 inhibitors (IL-12/23i), while none was identified for Janus Kinase Inhibitors (JAKi), Tocilizumab, Abatacept and Rituximab. We observed that microbiota and drugs are influenced in a mutual and reciprocal way. Indeed, microbiota seems to influence therapeutic response and efficacy, whereas in the other hand, drugs may restore healthy microbiota. (4) Conclusions: Future improvement in pharmacomicrobiomics could help to detect an effective biomarker able to guide treatment choice and optimize management of inflammatory arthritides.

## 1. Introduction

Inflammatory arthritides are a group of immune-mediated chronic diseases, including rheumatoid arthritis (RA), ankylosis spondylitis (AS) and psoriatic arthritis (PsA). Their pathogenesis is complex and still not clearly understood, even though genetic susceptibility, dysregulation of the immune system and environment effect seem to be involved [1,2]. Among non-genetic factors, increasing evidence suggest the role of gut and oral microbiota as a potential trigger of the inflammatory process [3,4,5,6,7].

The term “microbiota” is used to describe the set of microorganisms inhabiting gastroenteric tract, oral cavity, skin and genital area. The enteric microbiota is the most represented and includes from 30 to 400 trillion microorganisms whose composition is variable on the base of ethnicities, food intake and environmental agents. Nevertheless, microorganisms perform in a great number of host functions from digestion to metabolic processes [8,9]. Furthermore, the gut microbiota takes part in the development of immune system through a variety of mechanism: (a) promoting the establishment of gut-associated lymphoid tissue (GALT); (b) maintaining the balance between pro-inflammatory and anti-inflammatory agents; (c) favoring the tolerance of self-agents and (d) promoting the identification of pathogenic microorganism [6,7,8,10]. This interplay between immune system and microbiota is complex and it concerns both innate and adaptative response. A key role seems to be played by Short-Chain Fatty Acid (SCFA)-producing bacteria which through the generation of butyrate and propionate in the gut, might modulate the inflammatory process. A summary of the main mechanisms involved is reported in Figure 1. It seems that SCFAs are able to downregulate histone deacetylase (HDAC) and nuclear factor-ƘB (NF-ƘB) in a dose-dependent manner leading to a modification of gene expression which results in the differentiation of regulatory T cells (Tregs), in downregulation of lymphocytes T helper-17 (Th17) activity and in the increase production of anti-inflammatory cytokines, such as Interleukin-10 (IL-10) [11]. Moreover, recent evidence suggest that the GM may guide the differentiation of innate lymphoid cells (ILC) [12], a cluster of cells located in barrier tissues, able to maintain the barrier integrity through the production of various cytokines [13]. Additionally, the host microorganisms can stimulate the B cells to produce Immunoglobulins A (IgA) and promote CD4+ and CD8+ cells differentiation [14,15,16].

This mutual interplay between the GM and the immune response is supported by several studies [10,11,12,17,18]. The alteration of the microbiota composition is called “dysbiosis” and can act as a trigger for the development of autoimmune diseases, including Inflammatory Bowel Disease (IBD) and several inflammatory arthritides [19,20,21,22,23,24]. One of the most accredited theories concerns the molecular mimicry mechanism, in which cleaved microbial peptides can induce the stimulation of autoreactive T and B cells in genetically predisposed subjects [25,26,27,28,29]. Moreover, dysbiosis may induce a loss of balance between pro- and anti-inflammatory molecules leading to mucosal damage, thus promoting the microbial translocation [10,11]. All these processes determine the loss of Treg/Th17 balance and a pro-inflammatory environment.

Since the microbiota is implicated in the modulation of inflammatory cytokines, some authors speculated that GM might be a promising therapeutical target, as well as an effective biomarker able to predict therapy response, from the perspective of a personalized medicine approach [30].

Inflammatory arthritides have been treated for decades with conventional Disease Modifying Anti-Rheumatic drugs (cDMARDs) such as Methotrexate (MTX), Sulfasalazine and Leflunomide, obtaining a partial control of symptoms and inflammatory status. Since the advent of biological drugs (bDMARDs), from tumor necrosis factor alpha (TNF-α) inhibitors to the most recent Interleukin-17 inhibitors and Interleukin 12/23 inhibitors, the treatment of inflammatory arthritides has been revolutionized, allowing a better control of the disease activity [31,32,33,34]. Nevertheless, some patients with moderate or severe arthritis might require several attempts before the most effective drug is identified [16,17,18]: in this context, since biologics are expensive and patients get frustrated by treatment failure, researchers are trying to identify biomarkers able to predict therapeutic response.

Considering its influence on immunity homeostasis, the GM might affect the treatment response and nowadays there is an increasing attention to the “pharmacomicrobiomics”, in other words, the study of the dynamic interaction between GM and xenobiotics, putting emphasis on the role of microbiota in modifying drugs’ pharmacokinetics and pharmacodynamics. Viceversa, recent studies have shown that drugs can alter microbial composition, playing a reciprocal role [35].

Therefore, the individual variability in treatment response must be better understood, aiming at reducing failures and therapeutic switches in the framework of personalized medicine. The aim of our review was to highlight the current evidence on pharmacomicrobiomics applied to the main treatment of inflammatory arthritides, both cDMARDs and bDMARDs. Two main aspects of the complex interaction between microbiome and drugs are addressed: first, we explored the potential microbial biomarkers to predict therapeutic response, and second, we investigated how drugs are able to modify microbiota composition and function.

## 2. Materials and Methods

We performed a literature review on PubMed with the following MeSH terms: [microbiota OR microbiome] AND [inflammatory arthritides] AND [tumor necrosis factor inhibitors OR methotrexate OR interleukin-17 inhibitors OR interleukin 12–23 inhibitors OR tocilizumab OR Janus Kinase inhibitors OR abatacept OR rituximab]. Inclusion criteria for our research were: human clinical studies about microbiota in patients affected by Rheumatoid Arthritis, Psoriatic Arthritis and Ankylosis Spondylitis from any ethnicity and age, articles written in English and published until September 2022. We excluded articles written in other language than English and studies including patients with overlap conditions. PRISMA guidelines were not followed, given the narrative nature of our literature revision.

## 3. Results

This research retrieved 18 articles published from 2015 to 2022. Therapies provided from the latest ACR/EULAR recommendations on the management of RA, SA and PsA were analyzed in relation to microbiota alterations [36,37,38]. In detail, we focused on 10 manuscripts describing mainly MTX and TNF-α inhibitors, since a larger amount of evidence was retrieved for these compounds from a microbial point of view. Conversely, fewer data were available about IL-17i, while none was identified for IL-12/23i, Janus Kinase Inhibitors, Tocilizumab and Abatacept. Table 1 summarizes the selected papers and shows the mutual interaction between microbiota and drugs.

The main finding of our research was that microbiota may influence treatment response through different mechanisms such as the involvement of SCFA-producing bacteria and the balancing of pro- and anti-inflammatory cytokines. On the other hand, treatments may alter the aberrant microbial composition thus restoring the bacterial eubiosis.

### 3.1. Interaction of Methotrexate with Microbiome

MTX is a cDMARD largely used in the treatment of inflammatory arthritis [36], either in monotherapy or associated with biological drugs [49]. At present it is considered the first line cDMARD for most RA patients, despite up to 50% of them do not reach adequate clinical efficacy or experience adverse events [50,51]. Several factors influence the interindividual variability of response to MTX, such as age, sex, Body Mass Index (BMI), smoking status, genetics factors and some serum biomarkers [52].

It has been observed that MTX can influence the diversity of the various bacterial strains of the intestinal microbiota [53]. In addition, the microbiota may play a role in MTX gastrointestinal toxicity [54] and bioavailability.

#### 3.1.1. The Influence of Microbiota on MTX Response

To understand the complex interactions between MTX and microbiota it might be important to focus on the enzymes expressed by gut bacteria affecting MTX metabolism and bioavailability (Figure 2).

Methotrexate once into the cell is polyglutamated by folylpolyglutamate synthase (FPGS) to Methotrexate-polyglutamated (MTX-PGs), which is thought to be the more potent than MTX itself in inhibition of dihydrofolic reductase (DHFR) [55]. MTX-PG has a higher affinity for its target proteins than MTX, but also a lower affinity for folate transporters than MTX. This means that MTX-PGs are poorly transported in and out of cells. Removal of glutamate entities from MTX-PGs by glutamate carboxypeptidase 2 (CPDG2) reduces its efficacy as an inhibitor of DHFR. CPDG2 is an enzyme found in many gut bacteria, such as *Pseudomonas species*, *Streptococcus faecalis*, *Enterobacter aerogenes* and *Candida tropicalis* [56], so that they could play a role in altering efficacy of MTX. On the other hand, some intestinal bacteria have the capacity to add glutamate to MTX via FPGS-like enzymes. As MTX-PGs is poorly exported out the cell, it would affect treatment outcomes [57].

Effectively we found more studies concerning the role of microbiota in MTX- pharmacokinetics and pharmacodynamics. For example, Artacho et al. applied Machine Learning (ML) to the metagenomic data, aiming at developing a microbiome-based model to predict the lack of response to MTX in RA patients [39]. They studied GM before and after MTX administration in new onset, treatment-naive RA patients, comparing the bacterial strains of MTX-Responders (MTX-R) with those of non-responders (MTX-NR). MTX-R showed a significantly lower microbial diversity compared to MTX-NR. Bacteria in NR samples were: the *Euryarchaeotaphylum, unclassified Clostridiales/Clostridiales Incertae Sedis XIII* (family) and *Escherichia/Shigella*. By contrast, in MTX-R, *Prevotella* and *Bacteroides genus* were significantly more abundant.

In addition, Zhang X. et al. analyzed the oral and fecal microbiota of treatment-naïve RA patients before and after the use of MTX, in combination with other DMARDs [40]. Drug response was assessed through Disease Activity Score 28 (DAS28) and patients were stratified in three groups: good responders, moderate responders and poor responders. Interestingly, they found that the three groups differed for microbial composition at baseline. This study also generated predictive models using the microbiota data collected, which were able to differentiate between good and poor responders. The authors provided evidence that microbiota-based variables may be important in determining whether an RA patient might respond well or poorly to MTX, although the underlying mechanisms remain unknown, suggesting a potential diagnostic and prognostic value of GM in RA patients [40].

#### 3.1.2. The Influence of MTX on Microbiota Structure

On the other hand, many works highlighted that MTX could change microbiota composition, demonstrating the bidirectionality of this interaction. Particularly, in the aforementioned study by Zhang X. et al., they identified four unclassifiable taxa and one species, most closely related to *Enterococcus faecium*, whose abundance decreased after MTX. Despite this modification, they showed the achievement of eubiosis in the dental and salivary microbiota, rather than in the composition of fecal microbiota. Nayak et al. showed that MTX induced changes in GM and mitigate host immune responses in mice [56]. Specifically, they found that MTX was able to reduce the *Bacteroidetes phylum* in Germ-free (GF) mice that were colonized with stool samples from either healthy controls or RA patients. The authors selected 45 bacterial strains commonly found in the human intestine, documenting that *Bacteroides* was the most sensitive to MTX induced inhibition in culture compared to other phyla such as *Firmicutes, Actinobacteria, and Proteobacteria*. Moreover, the authors selected four bacterial isolates with varying sensitivity to MTX and performed RNA sequencing experiments. As expected, MTX induced a profound shift in gene expression in *Bacteriodes theta* (*B. theta*) that correlated with their sensitivity to MTX, while there was little gene shift in the MTX insensitive strains, such as *Clostridium sporogenes* and *Clostridium symbiosum*. Unexpectedly, a strong shift in gene expression was observed in one of the MTX insensitive strains, *C. asparagiforme*. Using metabolomics analyses, data showed that the MTX, as in mammalian cells, targets the purine and pyrimidine metabolic pathways in both *B. theta* and *C. asparagiforme*. Therefore, it appears that some bacteria strains such as *C. asparagiforme* can overcome the MTX inhibition of purine and pyrimidine metabolic pathways and be transcriptionally responsive to achieve an MTX-resistant status. Finally, they showed that GF mice colonized with the post MTX stool samples of RA patients displayed a reduction of some activated subset of T cells (in detail T helper 1 (Th1) cells and Th17 cells), myeloid cells, and B cells in spleen and/or intestinal mucosa when compared to pre-MTX stool samples, suggesting that the MTX-induced GM changes may decrease host immune activation.

Moreover, Funk and Becker tried to identify exogenous and microbiota-derived biomarkers of MTX efficacy in a prospective cohort of 30 children with JIA. The plasma samples collected before starting MTX and 3 months after were analyzed using a semi-targeted global metabolomic platform detecting 673 metabolites across a diversity of biochemical classes. The authors identified 50 metabolites which were significantly altered following the exposure to MTX. Reductions in three metabolites were found to be associated with clinical response measured by American College of Rheumatology Pediatric 70 (ACR Pedi 70) response criteria and represented several microbiota and exogenously derived metabolites including: dehydrocholic acid, biotin, and 4-picoline. Specifically, dehydrocholic acid is a secondary bile acid and is an oxidation product of cholic acid formed through enterohepatic recirculation of primary bile acids with metabolism occurring via the gut microbiota, so changes in dehydrocholic acid levels secondary to MTX therapy support the findings that MTX efficacy in autoimmune arthritis may be related to its effect on GM composition [41].

Finally, a multicenter study compared fecal microbiota of 45 treatment-naïve children with JIA with microbiota of 41 treated patients: 29 treated with MTX and 12 with etanercept. They did not find any significant difference in the GM structure and no significant differences in levels of fecal SCFAs before and during treatment with MTX or etanercept, suggesting that these changes were not related to their therapeutic effects [42].

### 3.2. Interaction of TNF-α Inhibitors with Microbiome

TNFi (TNF-α inhibitors) are still frequently represented as the first line bDMARDs used in RA, PsA and AS, as well as in many non-rheumatic diseases such as IBD, psoriasis, immune-mediated uveitis and hidradenitis suppurativa [33,58]. The TNFi group includes four monoclonal antibodies (mAbs) named Adalimumab, Certolizumab, Infliximab and Golimumab, and one soluble TNF-α receptor named Etanercept. Despite their great success in achieving remission or low diseases activity in a high percentage of treated patients, a variable proportion of patients experience primary inefficacy or secondary loss of response and no predictive factors of patient outcome have been identified [59].

#### 3.2.1. The Influence of Microbiota on TNF-α Inhibitors (TNFi) Response

Concerning the GM influence on TNFi response, Bazin et al. evaluated stool samples from 18 AS patients naïve to TNFi, at baseline (M0) and after 3 months (M3) from treatment initiation: 15 of them received etanercept, 2 adalimumab and 1 was treated with infliximab [43]. After 3 months, 8 patients were classified as non-responders given an Ankylosing Spondylitis Disease Activity Score (ASDAS) improvement ≤ 1. Then, they investigated the GM composition through 16S rRNA sequencing at M0 and M3 and compared it between responders and non-responders. The authors found that responders were characterized by a higher alpha-diversity than non-responders and higher microbiota composition stability at M3, whereas non-responders showed drastic changes. Furthermore, they found different markers depending on sample time. In particular, the enrichment of *Betaproteobacteria class* and *Burkholderiales order* at M0 were predictive of good response to TNFi, while at M3 different markers were observed. Moreover, *Dialister* was correlated with good response whereas the abundance of *Salmonella* was found in non-responders.

If these results are confirmed by more studies, it may pave the way to the development of predictive tests suitable for clinical practices to predict TNFi response.

#### 3.2.2. The Influence of TNFi on Microbiota Structure

Chen et al. compared 30 AS patients treated with Adalimumab with 24 HC in terms of fecal samples at baseline and after 6 months of therapy [44]. They identified that alpha-diversity in AS patients at baseline was lower than in HC, while Beta-diversity was higher before treatment and not statistically significant after therapy. These results suggest that TNFi might modify microbial community, although any difference or specific biomarkers in the microbial population was able to discriminate responders from non-responders. They noticed a higher abundance of *Comamonas genus* in non-responding patients, while no other bacteria genus was correlated to responsiveness. An explanation of the difference with the previous study could be related to the different ethnicities and dietary habits of the enrolled population.

Moreover, Yin et al. analyzed stool samples from 127 AS patients both before and after therapy with TNFi and compared them with those of 123 HC [45]. Eight bacterial species appeared to be differently present in pre-treatment patients and restored similarly to HC, after treatment. Species found depleted in pre-treatment patients were: *Prevotella copri, Faecalibacterium prausnitzii, Bilophila unclassified, Klebsiella pneumoniae, Ruminococcus bromii* and *Eubacterium biforme*. After therapy, all these species were restored to a composition similar to HC. The same happened to *Clostridium symbiosum* and *Eggerthella unclassified* which were enriched in pre-treatment patients and reduced to normal value after therapy. These data suggest that TNFi normalize gut dysbiosis regardless of drug response, with a not yet identified mechanism.

A similar study was conducted by Dai et al., since they compared the microbiota from 24 AS patients at baseline and after 1 month of therapy with those of 11 HC [46]. They identified a different microbial profile in pre- and post-treatment patients, with *Bacilli phylum* and *Haemophilus genus* being more abundant in pre-treatment patients, while post treatment patients were enriched with *Megamonas* and *Lachnoclostridium genus*. Interestingly, the latter two were found abundant also in HC, confirming that TNFi can restore microbial composition even after a short-term therapy. Additionally, they also established that *Megamonas* and *Lachnoclostridium genus* were negatively correlated with disease activity calculated with the BASDAI, while *Haemophilus* correlated positively. In particular, it seems that *Megamonas* could reduce TNF-alpha level in a dose-dependent method.

Other studies tried to identify predictive markers of disease activity. Zhan et al. analyzed the stool samples from 19 HC and 20 patients with spondyloarthritis (both AS and PsA) at Adalimumab treatment initiation and after 1, 3 and 6 months, aiming to identify dynamic GM variation [47]. The disease activity was assessed by BASDAI and Simplified Psoriasis Index (SPI). BASDAI positively correlated with the abundance of *Escherichia-Shigella* and *Klebsiella*, and negatively with *Lachnospiraceae*. Moreover, they noticed that bacteria composition fluctuated within 6 months of therapy. *Bifidobacterium* and *Parasutterella* rise to normal value after therapy while *Escherichia-Shigella* and *Klebsiella* decreased to normal value. Other species variated during treatment with less regularity.

These studies show that GM in AS patients is different from that of healthy subjects and treatment with TNFi re-estabilished the eubiosis with the increase of those bacterial strains found in HC.

### 3.3. Interaction of IL-17 Inhibitors with Microbiome

The IL-17i group, including secukinumab and ixekizumab, are monoclonal antibodies used for treatment of PsA/SpA and plaque psoriasis. IL-17 plays a role in maintain epithelial health and this could be why IL-17i have been associated with Crohn’s disease (CD) exacerbation and candidiasis.

Considering their recent introduction, few studies are available to date regarding their effect on microbiota and they concerned mostly on IL-17i induced gut inflammation in PsA patients.

#### The Influence of IL-17 Inhibitors on Microbiota Structure

Manasson et al. first studied the effects of the IL-17A inhibition on gut bacterial and fungal communities [48]. Fecal samples from PsA/SpA patients pre- and post-treatment with TNFi or an anti-interleukin (IL)-17A monoclonal antibody inhibitor (IL-17i; *n* = 14) underwent sequencing and computational microbiota analysis. The fecal levels of fatty acid metabolites and inflammatory cytokines or intestinal inflammation were correlated with sequence data. Then, ileal biopsies obtained from SpA patients who developed clinically overt Crohn’s disease after treatment with IL-17i (*n* = 5) were analyzed for expression of IL-23/Th-17 related cytokines, IL-25/IL-17E-producing cells and type-2 innate lymphoid cells (ILC2s). After treatment with IL-17i, there were significant shifts in abundance of specific taxa particularly *Clostridiales* (*p* = 0.016) and *Candida albicans* (*p* = 0.041), compared to TNFi.

Ileal biopsies showed that clinically overt Crohn’s disease was associated with expansion of IL-25/IL-17E-producing tuft cells and ILC2s compared to pre-IL-17i treatment levels. In this study, the IL-17A blockade correlated to subclinical gut inflammation and intestinal dysbiosis of some bacterial and fungal taxa, such as *C. albicans*. Furthermore, IL-17i-related Crohn’s disease is associated with overexpression of IL-25/IL-17E-producing tuft cells and ILC2s. These alterations may explain the link between the inhibition of IL-17 pathway and the (sub)clinical gut inflammation observed in SpA [48].

## 4. Discussion

Our narrative review explores the complex interaction between drugs used in inflammatory arthritides and the gut microbiota, highlighting the potential utility of the application of pharmacomicrobiomics to a personalized medicine approach. Treatment selection for the single patient is still a daily challenge, as many factors play a role in determining the effective therapeutic response. Indeed, it is known that a variable proportion of patients do not achieve remission nor low disease activity, causing multiple therapeutic switches and increasing patients’ frustration and health care cost. The potential role of pharmacomicrobiomics in rheumatology borrows concepts from oncology studies, in which the influence of gut microbiome on chemotherapy and immunotherapy has been proved [60,61].

Our data show that HC and patients with inflammatory arthiritides present different GM compositions with SCFA-producing bacteria, being more abundant in the former. It is well documented that the SCFAs are fundamental for the maintenance of immune homeostasis, playing a role in the differentiation of Treg cells and increasing the production of anti-inflammatory cytokines. Therefore, modifications in their abundance may affect the development and the maintenance of an inflammatory status, typical of rheumatic disease. This bacterial variation has also been proved in different cohorts of patients suffering from other conditions, such as cancer, IBD, psoriasis and immune-mediated uveitis [62], even though the difference in microbial composition was not reproducible in the different studies, suggesting that the GM structure is influenced by multiple factors such as ethnicity and food intake.

Moreover, the clarification of the pharmacomicrobiomic interactions may pave the way to the identification of potential biomarkers to predict the therapeutic responses. To this purpose, few articles focused on GM diversity between responders and non-responders, providing contrasting results. In fact, it has been observed that an enrichment of *Betaproteobacteria class* and *Burkholderiales order* in treatment-naïve patients could be predictive of a good response to TNF-inhibitors, while the enrichment of *Salmonella* was associated with failure [43]. Other studies on Secukinumab and Ustekinumab highlighted the difference between responder and non-responders with a higher relative abundance of *Citrobacter, Staphylococcus*, and *Hafnia/Obesumbacteriuin* in the former group [63].

These results are in agreement with other oncology studies: indeed, responders to cancer immunotherapy are characterized by a different GM signature [61]. These signatures were associated with enhanced systemic immunity via a number of mechanisms, including the interaction of microbial components with antigen-presenting cells (APCs) and innate effectors (via Toll-like receptors), the induction of cytokine production by APCs or lymphocytes, and even local or distant effects of microbial metabolites.

Lastly, some treatments (both MTX and TNFα inhibitors) may revert dysbiosis to eubiosis, independently from the clinical response. Multiple studies proved that therapies could increase the abundance of SCFAs producing bacteria, such as *Megamonas* and *Lachnoclostridium* [64]. This data has been confirmed also in studies concerning other autoimmune diseases such as Crohn’s Disease and Ulcerative Colitis, in which there was a complete restoration of normal microbiota even after short time treatment [65,66,67,68]. Considering these results, the microbiome could be a target for the regulation of inflammatory autoimmune diseases through different modalities, such as by using prebiotics or probiotics or with a dietary intervention as well as fecal microbiota transplantation (FMT) or phage therapy. These approaches have already shown good results in mice models [69], but also few cases of FMT from a healthy donor to patients with drugs-refractive immune colitis have been reported, leading to restoration of the GM and of the proportion of regulatory T cells, as well as to the improvement of symptoms and clinical condition [70].

## 5. Conclusions

In summary, inflammatory arthritides are characterized by gut dysbiosis that can be restored by treatments. On the other hand, the microbiota seems to influence treatment response and drug effectiveness. In the future, promising methods for the GM modulation could arise, such as the FMT or personalized phage therapy. In this framework, studies on the GM shaping and on pharmacomicrobiomics might bring important advances in the management of rheumatic diseases, guiding the physician to tailor a personalized treatment, optimizing disease management and trying to limit drug failure.

## Figures and Tables

**Figure 1 genes-14-00089-f001:**
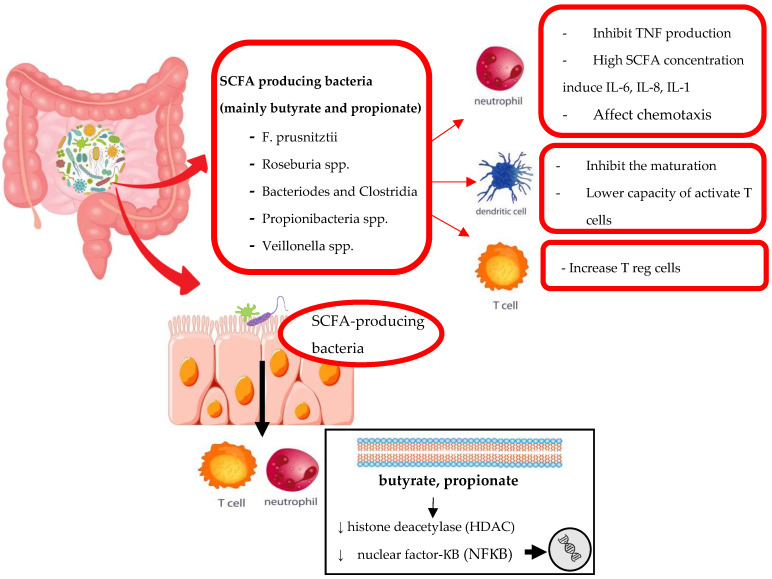
SCFA-producing bacteria generate butyrate, propionate and acetate in the gut. These compounds are able to inhibit histone deacetylase and nuclear factor-ƘB provoking a modification in gene expression. Through this mechanism, SCFAs promote T reg cells differentiation, downregulate Th17 activity and induce the production of anti-inflammatory cytokines such as IL-10. On the other hand, very high concentration of SCFAs may induce the production of IL-1, IL-6 and IL-8. An imbalance in SCFA-producing bacteria might be the cause of inflammatory status.

**Figure 2 genes-14-00089-f002:**
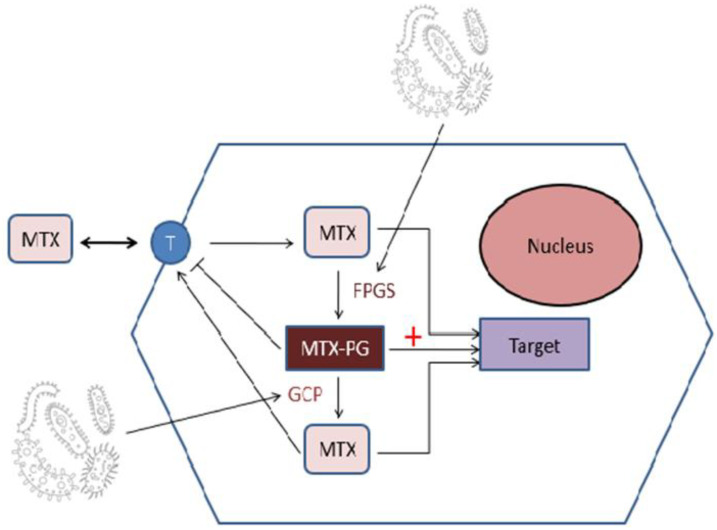
Once into cell, MTX is polyglutamated to MTX-PG by the enzyme folylpolyglutamate synthase (FPGS). This compound gains more affinity for the drug target, but it loses affinity for MTX-Transporter (T) hindering the release from the cell. Another enzyme, named glutamate carboxypeptidase (GCP), removes glutamate from MTX-PG permitting the transportation out of the cell. FPGS and GCP are enzymes expressed by many gut bacteria, suggesting a potential role in the efficacy of MTX.

**Table 1 genes-14-00089-t001:** Analyzed studies on the interaction between microbiome and therapies provided from the latest ACR/EULAR recommendations on the management of RA, SA and PsA. PT: patients, HC: Healthy Controls, RA: Rheumatoid Arthritis, MTX: Methotrexate, R: Responders, NR: Non-Responders, JIA: juvenile idiopathic arthritis, ETN: Etanercept, ADA: Adalimumab, INF: Infliximab, AS: Ankylosis Spondylitis, SpA: Spondyloarthritis, PsA: Psoriatic Arthritis, BASDAI: Bath Ankylosing Spondylitis Disease Activity Index, TNFi: Tumor Necrosis Factor Inhibitors, IL-17i: Interleukin-17 Inhibitors.

**Study**	**N° PT**	**N° HC**	**Disease**	**Therapy**	**Sample**	**Results**
Artacho, 2021 [39]	26	21	RA	MTX	Stool	R showed lower microbiota diversity compared to NR NR were enriched with *Euryarchaeotaphylum unclassified*, *Clostridiales/Clostridiales Incertae Sedis XIII* (*family*) and *Escherichia/Shigella*. R enriched with *Prevotella* and *Bacteroides*
Zhang X., 2015 [40]	77	80	RA	MTX	Stool	Restoration of microbiota after MTX. Variation in oral microbiota after treatment.
Funk and Becker, 2021 [41]	30	0	JIA	MTX	Plasma	Metabolomic analysis identified 50 metabolites that were significantly altered following the initiation of MTX. Reduction of 3 metabolites representing several microbiota and exogenously derived metabolites were found in R: dehydrocholic acid, biotin and 4-picoline.
Öman, 2021 [42]	41	45	JIA	29 MTX 12 ETN	Stool	No microbiota variation after both MTX and ETN.
Bazin, 2018 [43]	18	0	AS	15 ETN 2 ADA 1 INF	Stool	R higher alpha-diversity compared to NR and higher microbiota stability after treatment. NR had drastic change in microbial composition. *Betaproteobacteria class* and *Burkholderiales order* found in stool sample before treatment were predictive of good response. Enrichment of *Dialister* was found in R patients after therapy NR were enriched with *Salmonella*
Chen, 2021 [44]	30	24	AS	ADA	Stool	Alpha-diversity at baseline was lower in patients than in HC. Beta-diversity was higher in AS and did not change after therapy. No biomarker predictive of therapy response was found Higher abundance of *Comamonas genus* in NR
Yin, 2020 [45]	127	123	AS	TNFi	Stool	Patients were depleted of *Prevotella copri*, *Faecalibacterium prausnitzii, Bilophila unclassified, Klebsiella pneumoniae, Ruminococcus bromii* and *Eubacterium biforme,* compared with HC. Patients were enriched with *Clostridium symbiosum* and *Eggerthella unclassified*, compared to HC. Differences in microbiota composition was restored after treatment to levels of HC.
Dai, 2022 [46]	24	11	AS	TNFi	Stool	Pre-treatment patients were enriched with *Bacilli phylum* and *Haemophilus genus*. Post treatment patients as well as HC were enriched with *Megamonas* and *Lachnoclostridium genus*. *Megamonas* and *Lachnoclostridium* genus were negatively correlated with BASDAI, while *Haemophilus* correlated positively. *Megamonas* was found depleted in pre-treatment patients and restored after treatment. It seems to be involved in the reduction of TNFα level in a dose-depending manner.
Zhang, 2020 [47]	20	19	SpA (both AS and PsA)	ADA	Stool	BASDAI was positively correlated with the abundance of *Escherichia-Shigella* and *Klebsiella,* and negatively with *Lachnospiraceae*. Restoration of microbiota: *Bifidobacterium* and *Parasutterella* rose to normal value after therapy while *Escherichia-Shigella* and *Klebsiella* decreased to normal value.
Manasson, 2020 [48]	29	0	PsA/AS	15 TNFi 14 IL-17i	Stool (*n* = 29), Ileal biopsy (*n* = 5)	Patients treated with IL-17i were enriched with *Clostridiales* and *C. Albicans* compared to TNFi-treated.

## Data Availability

Not applicable.

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
