# Peer review of "The Yin-Yang Pharmacomicrobiomics on Treatment Response in Inflammatory Arthritides: A Narrative Review"

_genes, 2022, doi:10.3390/genes14010089_

Round 1

Reviewer 1 Report

The presented manuscript is a narrative review on inflammation arthritides. The comments are as below:

1. inflammatory arthritides and rheumatic disease are used interchangeably. suggest to standardize the use of this two terms.

2. What is the inclusion and exclusion criteria?

3. What is the articles selection process? reviewed by two authors?

4. PRISMA flowcharts is not in the manuscript.

5. The description of table 1 is too long. Please summarize the important points.

6. Section 3 were described in details the findings of studies included. This section is lengthy and what is the main findings the authors want to highlight? Merely reports the findings of the journals is insufficient.

7. Please throughly check the manuscripts for syntax errors. Eg, Clostridium sporogenes not italicized in the manuscript.

Author Response

Dear Editor, 

Thank you very much for your precious advices. We modified the article according to your suggestions as follow: 

  1. inflammatory arthritides and rheumatic disease are used interchangeably. suggest to standardize the use of this two terms.

Thank you for this advice. Effectively we chose to use the only the term 'inflmmatory arthritides' since our work is focus on this topic, in order to avoid any misunderstanding.

2. What is the inclusion and exclusion criteria? 

Thank you very much. We included the following sentences in order to clarify our selection process: 

Inclusion criteria for our research were: human clinical studies about microbiota in patients affected by Rheumatoid Arthritis, Psoriatic Arthritis and Ankylosis Spondylitis from any ethnicity and age, articles written in English and published until September 2022. We excluded articles written in other language than English and studies including patients with overlap conditions.

3. What is the articles selection process? reviewed by two authors?

Since the nature of our work is a narrative review, actually we didn't follow the PRISMA guidilines. Selection process was realized according the inclusion and exclusion criteria by the two Authors. Actually we didn't modified this part in the article, but let us know if you need further clarifications.

4. PRISMA flowcharts is not in the manuscript.

Our work is a narrative review, therefore we didn't follow the PRISMA guidelines. On matherial and Methods section we wrote the following sentence to clarify: PRISMA guidelines were not followed, given the narrative nature of our literature revision.

5. The description of table 1 is too long. Please summarize the important points.

We summerized in just one phrase: Analyzed studies on the interaction between microbiome and therapies provided from the latest ACR/EULAR recommendations on the management of RA, SA and PsA. The lenghty of the caption is given by the acronyms explanation. 

6. Section 3 were described in details the findings of studies included. This section is lengthy and what is the main findings the authors want to highlight? Merely reports the findings of the journals is insufficient.

Thank you very much for the suggestion. We reorganized the work adding some explanation about pharmacokinetic because we think it would be important to understand our aim, in particular we wrote these sentences supported by an explication image: 

Methotrexate (MTX) once into the cell is polyglutamated by folylpolyglutamate synthase (FPGS) to MTX-PG, which is thought to be the more potent than MTX itself in inhibition of dihydrofolic reductase (DHFR). MTX-PG has a higher affinity for its target proteins than MTX, but also a lower affinity for folate transporters than MTX. This means that MTX-PGs are poorly transported in and out of cells. Removal of glutamate entities from MTX-PGs by carboxypeptidase reduces its efficacy as an inhibitor of DHFR. Glutamate carboxypeptidase 2 (CPDG2) is an enzyme found in many gut bacteria, such as Pseudomonas species, Streptococcus faecalis, Enterobacter aerogenes and Candida tropicalis [55], so that they could play a role in altering efficacy of MTX. On the other hand, some intestinal bacteria have the capacity to add glutamate to MTX via FPGS-like enzymes. As MTX-PG is poorly exported out the cell, it would affect treatment outcomes.

We also added this phrase to specify the aim of our interest in this review.  We added the following sentence at the end of paragraph 3.2.1: 

If these results are confirmed by more studies, it may pave the way to the development of predictive tests suitable for clinical practices to predict TNFi response.

and at the end of paragraph 3.2.2: 

These studies show that GM in AS patients is different from that of healthy subjects and treatment with TNFi re-estabilished the eubiosis with the increase of those bacterial strains found in HC.

7. Please throughly check the manuscripts for syntax errors. Eg, Clostridium sporogenes not italicized in the manuscript.

Thank you very much, we are very sorry for the errors. We checked for syntax errors and we wrote all the bacteria in italicized letters.

Reviewer 2 Report

In this review, the authors were interested in the interactions between Disease Modifying Anti-Rheumatic Drugs and the gut microbiota. Although the manuscript is rich in interesting informations, I have some comments that I would like the authors to take into consideration. 

1- Firstly, I have the impression that the manuscript constitutes a synthesis of bacterial species that decrease and others that increase in the presence of a given drug without a deep explanation on the impact on the pharmacokinetics or pharmacodynamics of the drug concerned. Also, I notice that sometimes there are parts on the effect of the drug on the gut microbiota included in the section "effect of the microbiota on the drug".

Thus, an additional writing effort, and organization of ideas is required

2- Secondly, I would have liked to see illustrative figures, allowing to simplify and better understand the different microbiota-drug interactions. 

3- Please check the following sentence: 

Families of Proteobacteria, Pseudomonadaceae, Enterobacteriaceae and the order of Pseudomonadales increased significantly following secukinumab therapy, while Bacteroidetes and Firmicutes decreased after ustekinumab treatment.  

Wouldn't the decrease of Bacteroidetes and Firmicutes be related to secukinumab treatment instead of ustekinumab? 

4-Finally I draw the attention of the authors to some spelling mistakes: 

ex : 

ot : page 9, line 307

Obesumbacteriu page 9 line 310

and to put the meaning of the abbreviations at their first appearance, ex cDMARD and bDMARD in the abstract.

Author Response

Dear Editor, 

Thank you very much for your precious advices. We modified the article according to your suggestions as follow: 

1- Firstly, I have the impression that the manuscript constitutes a synthesis of bacterial species that decrease and others that increase in the presence of a given drug without a deep explanation on the impact on the pharmacokinetics or pharmacodynamics of the drug concerned. Also, I notice that sometimes there are parts on the effect of the drug on the gut microbiota included in the section "effect of the microbiota on the drug".

As you suggested, we added information about pharmocokinetics of the drug concerned. In particular we added the following sentences, also supported by an image: 

Methotrexate once into the cell is polyglutamated by folylpolyglutamate synthase (FPGS) to Methotrexate-polyglutamated (MTX-PGs), which is thought to be the more potent than MTX itself in inhibition of dihydrofolic reductase (DHFR). MTX-PG has a higher affinity for its target proteins than MTX, but also a lower affinity for folate transporters than MTX. This means that MTX-PGs are poorly transported in and out of cells. Removal of glutamate entities from MTX-PGs by carboxypeptidase reduces its efficacy as an inhibitor of DHFR. Glutamate carboxypeptidase 2 (CPDG2) is an enzyme found in many gut bacteria, such as Pseudomonas species, Streptococcus faecalis, Enterobacter aerogenes and Candida tropicalis [55], so that they could play a role in altering efficacy of MTX. On the other hand, some intestinal bacteria have the capacity to add glutamate to MTX via FPGS-like enzymes. As MTX-PGs is poorly exported out the cell, it would affect treatment outcomes.

Moreover,effectivly some part of the effect of gut microbiota were included in the other section, to this purpose we reorganized the section. We shifted the following phrase from paragraph 3.1.1 to paragraph 3.1.2

'Particularly, in the aforementioned study by Zhang X. et al, they identified four unclassifiable taxa and one species, most closely related to Enterococcus faecium, whose abundance decreased after MTX. Despite this modification, they showed the achievement of eubiosis in the dental and salivary microbiota, rather than in the composition of fecal microbiota.'

2- Secondly, I would have liked to see illustrative figures, allowing to simplify and better understand the different microbiota-drug interactions. 

We created two figures: one concerning the role of SCFA producing bacteria in determing inflammatory arthritides and other concerning the pharmacokinetics of MTX.

3- Please check the following sentence: 

Families of Proteobacteria, Pseudomonadaceae, Enterobacteriaceae and the order of Pseudomonadales increased significantly following secukinumab therapy, while Bacteroidetes and Firmicutes decreased after ustekinumab treatment.  Wouldn't the decrease of Bacteroidetes and Firmicutes be related to secukinumab treatment instead of ustekinumab? 

Regarding this sentence: actually we decided to take it off because it was confusing and we reorganized the concept as follow: 

Fecal samples from PsA/SpA patients pre- and post-treatment with TNFi or an anti-interleukin (IL)-17A monoclonal antibody inhibitor (IL-17i; n=14) underwent sequencing and computational microbiota analysis. The fecal levels of fatty acid metabolites and inflammatory cytokines or intestinal inflammation were correlated with sequence data. Then ileal biopsies obtained from SpA patients who developed clinically overt Crohn’s disease after treatment with IL-17i (n=5) were analyzed for expression of IL-23/Th-17 related cytokines, IL-25/IL-17E-producing cells and type-2 innate lymphoid cells (ILC2s). After treatment with IL-17i, there were significant shifts in abundance of specific taxa particularly Clostridiales (p=0.016) and Candida albicans (p=0.041), compared to TNFi.

4-Finally I draw the attention of the authors to some spelling mistakes: 

ex : 

ot : page 9, line 307

Obesumbacteriu page 9 line 310

and to put the meaning of the abbreviations at their first appearance, ex cDMARD and bDMARD in the abstract.

Thank you very much for the report, we are very sorry for the errors. We revised the aformentioned errors and as well other errors present in the text. Moreover we put the meaning of abbreviation in their very first apearance.

Round 2

Reviewer 2 Report

I thank the authors for their response to my first question. On the other hand, they cited for the added part, reference 55 which corresponds to the article "Association of anti-TNF-α treatment with gut microbiota of patients with ankylosing spondylitis". This article deals with anti-TNF-α and not methotrexate.

About the references:

The reference numbers that appear in Table 1 do not correspond to those found in the references section.

Also, the form used to mention some references is not homogeneous with the others. Example: references N° 43-46 start with the title and not with the name of the author.

Concerning my remark on the illustrative figures, I thank the authors for taking this remark into consideration. On the other hand, I think that figure 1 is not completely finished seeing the words underlined in red

Regarding my question about ustekinumab and secukinumab. I just asked the author to check if there was an error. I didn't understand why the authors deleted this part. 

Author Response

Dear Editor, 

Thank you very much for your revision. We modified the article as follows: 

1) I thank the authors for their response to my first question. On the other hand, they cited for the added part, reference 55 which corresponds to the article "Association of anti-TNF-α treatment with gut microbiota of patients with ankylosing spondylitis". This article deals with anti-TNF-α and not methotrexate.

About the references:

The reference numbers that appear in Table 1 do not correspond to those found in the references section. Also, the form used to mention some references is not homogeneous with the others. Example: references N° 43-46 start with the title and not with the name of the author.

Thank you very much to have marked the bibliography errors. Unfortunatly there was an error that messed up all the subsequent references. Once found the problem, we checked all the bibliography and we made it homogenous. At present, It should be correct. 

2)Concerning my remark on the illustrative figures, I thank the authors for taking this remark into consideration. On the other hand, I think that figure 1 is not completely finished seeing the words underlined in red.

Thank you very much for the tip. Of course we modified the figures in order to remove the red signs. 

3)Regarding my question about ustekinumab and secukinumab. I just asked the author to check if there was an error. I didn't understand why the authors deleted this part. 

Regarding this part, we actually decided to remove the phrase because it concerned an article about psoriasis with few allusions to psoriatic arthritis. Since our research was tightly about  arthritides, we did not consider to be relevant in the the results part. Anyway, we mentioned it on the Discussion part, as follows: 

'Other studies on Secukinumab and Ustekinumab highlighted the difference between responder and non-responders with a higher relative abundance of Citrobacter, Staphylococcus, and Hafnia/Obesumbacteriuin in the former group'

Thank you very much for your work

Best Regards

the Authors

Round 3

Reviewer 2 Report

I thank the authors for taking my comments into consideration.

I have no further questions